# Context-Aware Synthesis and Placement of Object Instances

**Donghoon Lee**[1,2][*],    **Sifei Liu**[3],    **Jinwei Gu**[3],    **Ming-Yu Liu**[3],
**Ming-Hsuan Yang**[2,4],    **Jan Kautz**[3]

donghoon.lee@rllab.snu.ac.kr   {sifeil, jinweig, mingyul}@nvidia.com
mhyang@ucmerced.edu   jkautz@nvidia.com

[1]Seoul National University, [2]Google Cloud AI, [3]NVIDIA, [4]University of California at Merced

## Abstract

Learning to insert an object instance into an image in a semantically coherent manner is a challenging and interesting problem. Solving it requires (a) determining a location to place an object in the scene and (b) determining its appearance at the location. Such an object insertion model can potentially facilitate numerous image editing and scene parsing applications. In this paper, we propose an end-to-end trainable neural network for the task of inserting an object instance mask of a specified class into the semantic label map of an image. Our network consists of two generative modules where one determines *where* the inserted object mask should be (i.e., location and scale) and the other determines *what* the object mask shape (and pose) should look like. The two modules are connected together via a spatial transformation network and jointly trained. We devise a learning procedure that leverage both supervised and unsupervised data and show our model can insert an object at diverse locations with various appearances. We conduct extensive experimental validations with comparisons to strong baselines to verify the effectiveness of the proposed network. Code is available at https://github.com/NVlabs/Instance_Insertion.

## 1   Introduction

Inserting objects into an image that conforms to scene semantics is a challenging and interesting task. The task is closely related to many real-world applications, including image synthesis, AR and VR content editing and domain randomization [28]. Numerous methods [2, 7, 8, 10, 15, 16, 20, 24, 26, 31, 33–35] have recently been proposed to generate realistic images based on generative adversarial networks (GANs) [5]. These methods, however, have not yet considered semantic constraints between scene context and the object instances to be inserted. As shown in Figure 1, given an input semantic map of a street scene, the context (i.e., the road, sky, and buildings) constrains possible locations, sizes, shapes, and poses of pedestrians, cars, and other objects in this particular scene. Is it possible to learn this conditional probabilistic distribution of object instances in order to generate novel semantic maps? In this paper, we propose a conditional GAN framework for the task. Our generator learns to predict plausible locations to insert object instances into the input semantic label map and also generate object instance masks with semantically coherent scales, poses and shapes. One closely related work is the ST-GAN approach [14] which warps an input RGB image of an object segment to place it in a scene image. Our work differs from the ST-GAN work in two ways. First, our algorithm operates in the semantic label map space. It allows us to manipulate the scene without relying on the RGB image that needs to be rendered in a number of cases, e.g., simulators or virtual worlds. Second,

---

[*]This work was predominantly conducted when Donghoon Lee was an intern at NVIDIA.

we aim to learn the distribution of not only locations but also shapes of objects conditioned on the input semantic label map.

Our conditional GANs consist of two modules tailored to address the *where* and *what* problems, respectively, in learning the distributions of object locations and shapes. For each module, we encode the corresponding distributions through a variational auto-encoder (VAE) [11, 23] that follows a unit Gaussian distribution in order to introduce sufficient variations of locations and shapes.

The main technical novelty of this work is to construct an end-to-end trainable neural network that can sample plausible locations and shapes for the new object from its joint distribution conditioned on the input semantic label map. To achieve this goal, we establish a differentiable link, based on the spatial transformation network (STN) [9], between two modules that are designed specifically for predicting locations and generating object shapes. In the forward pass, the spatial transformation network is responsible for transforming both of a bounding box for the *where* module and a segmentation mask for the *what* module to the same location on the input semantic label map. In the backward pass, since the spatial transformation network is differentiable, the encoders can receive the supervision signal that are back-propagated from both ends, e.g., the encoder of the *where* module is adjusted according to the discriminator on top of the *what* module. The two modules can benefit each other and be jointly optimized. In addition, we devise a learning procedure that consists of a supervised path and an unsupervised path to alleviate mode collapse while training. During inference, we use only the unsupervised path. Experimental results on benchmark datasets show that the proposed algorithm can synthesize plausible and diverse pairs of location and appearance for new objects.

The main contributions of this work are summarized as follows:

- We present a novel and flexible solution to synthesize and place object instances in images, with a focus on semantic label maps. The synthesized object instances can be used either as an input for GAN-based methods [8, 30, 31, 34] or for retrieving the closest segment from an existing dataset [19], to generate new images.

- We propose a network that can simultaneously model the distribution of where and what with respect to an inserted object. This framework enables both modules to communicate and optimize each other.

## 2 Related Work

The key issues of inserting object instances to images are to consider: (a) where to generate an instance, and (b) what scale and shape, or pose is plausible, given a semantic mask. Both are fundamental vision and learning problems that have attracted much attention in recent years. In this section, we discuss methods closely related to this work.

**Predicting instance locations.** It mainly concerns with the geometric consistency between source and target images, which falls into the boarder category of scene and object structure reasoning [1, 4, 32]. In [29], objects are detected based on the contextual information by modeling the statistics of low-level features of the object and surrounding scene. The same problem is recently addressed using a deep convolutional network [25]. Both methods predict whether objects of interest are likely to appear at specific locations without determining additional information such as scale. In [27], an image composition method is developed where the bounding box position of an object is predicted based on object proposals and a new instance is retrieved from an existing database. However, this framework makes it significantly challenging to develop a joint generation model of shape and position since the objective function with respect to object location prediction is not differentiable. Recently, the ST-GAN [14] utilizes spatial transformer network to insert objects based on image warping. However, this method does not synthesize plausible object instances in images.

**Synthesizing object instances.** Object generation based on GANs has been studied extensively [2, 7, 8, 10, 15, 16, 20, 24, 26, 31, 33–35]. Closest to this work are the methods designed to in-paint a desired patch [18] or object (e.g., facial component [13], pedestrian [17]) in original images. In contrast, the object generation pipeline of this work is on the semantic layout rather than image domain. As a result, we simplify the network module on learning desirable object shapes.

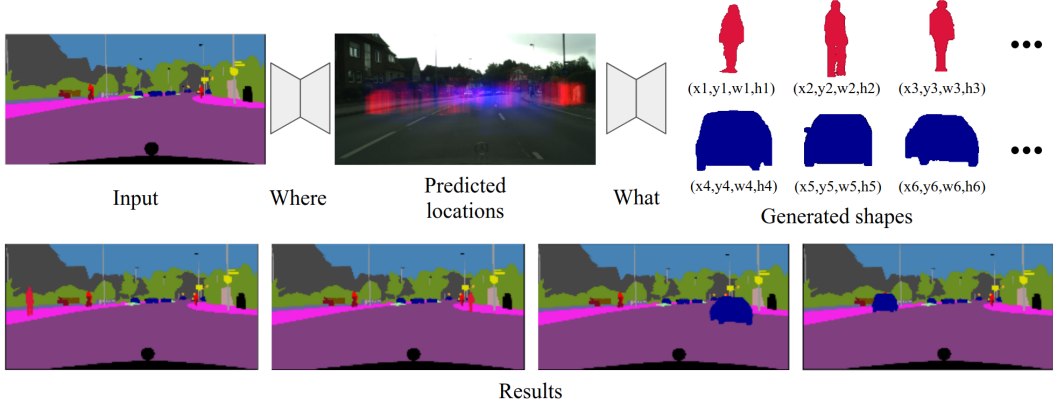

Figure 1: Overview of the proposed algorithm. Given an input semantic map, our end-to-end trainable network employs two generative modules, i.e., the *where* module and the *what* module, in order to learn the spatial distribution and the shape distribution of object instances, respectively. By considering the scene context, the proposed algorithm can generate multiple new semantic maps by synthesizing and placing new object instances at valid locations with plausible shape, pose, and scale.

**Joint modeling of what and where.** Several methods have been developed to model what and where factors in different tasks. Reed et al. [22] present a method to generate an image from the text input, which can either be conditioned on the given bounding boxes or keypoints. In [32], Wang et al. model the distribution of possible human poses using VAE at different locations of indoor scenes. However, both methods do not deal with the distribution of possible location conditioned on the objects in the scene. Hong et al. [6] model the object location, scale and shape to generate a new image from a single text input. Different from the proposed approach, none of the existing methods are constructed based on end-to-end trainable networks, in which location prediction, as well as object generation, can be regularized and optimized jointly.

# 3 Approach

The proposed algorithm learns to place and synthesize a new instance of a specific object category (e.g., car and pedestrian) into a semantic map. The overall flow is shown in Figure 1.

Given the semantic map as an input, our model first predicts possible locations where an object instance is likely to appear (see the *where* module in Figure 1 and Figure 2). This is achieved by learning affine transformations with a spatial transformer network that transforms and places a unit bounding box at a plausible location within the input semantic map. Then, given the context from the input semantic map and the predicted locations from the *where* module, we predict plausible shapes of the object instance with the *what* module (see Figure 1 and Figure 3). Finally, with the affine transformation learned from the STN, the synthesized object instance is placed into the input semantic map as the final output.

As both the *where* module and the *what* module aim to learn the distributions of the location and the shape of object instances conditioned on the input semantic map, they are both generative models implemented with GANs. To reduce mode collapse, during training, both the *where* module and the *what* module consist of two parallel paths — a supervised path and an unsupervised path, as shown in Figure 2 and 3. During inference, only the unsupervised path is used. We denote the input to the unsupervised path as $\mathbf{x}$, which consists of a semantic label map and an instance edge map that can be extracted from the dataset (we use datasets that provide both semantic and instance-level annotations, e.g., Cityscapes [3]). In addition, we use $\mathbf{x}^{+}$ to denote the input to the supervised path, which also consists of a semantic label map and an instance edge map, but also contains at least one instance in the target category. Table 1 describes the symbols used in our approach. In the following, we describe the two generators and four discriminators of the proposed method.

Table 1: Symbols used in our approach. We have two generators and four discriminators in total.

| Symbol | Description | Symbol | Description |
|---|---|---|---|
| $G_l$ | generator (instance location) | $G_s$ | generator (instance shape) |
| $D_l$ | $(D_{layout}^{box}, D_{affine})$ | $D_s$ | $(D_{layout}^{instance}, D_{shape})$ |
| $D_{layout}^{box}$ | discriminator (semantic map w/ bbox) | $D_{layout}^{instance}$ | discriminator (semantic map w/ instance) |
| $D_{affine}$ | discriminator (affine transform) | $D_{shape}$ | discriminator (instance shape) |
| $\mathcal{L}_l(G_l, D_l)$ | loss for location prediction (1) | $\mathcal{L}_s(G_s, D_s)$ | loss for shape prediction (5) |

## 3.1 The *where* module: learning a spatial distribution of object instances

As shown in Figure 2, given the input semantic map $\mathbf{x}$, the *where* module aims to learn the conditional distribution of the location and size of object instances valid for the given scene context. We represent such spatial (and size) variations of object instances with affine transformations of a unit bounding box $b$. Thus, the *where* module is a conditional GAN, where the generator $G_l$ takes $\mathbf{x}$ and a random vector $z_l$ as input and outputs an affine transformation matrix $A$ via a STN, i.e., $G_l(\mathbf{x}, z_l) = A$. We denote $A(obj)$ as applying transformation $A$ to the $obj$.

We represent a candidate region for a new object by transforming a unit bounding box $b$ into the input semantic map $\mathbf{x}$. Without loss of generality, since all objects in training data can be covered by a bounding box, we can constrain the transform as an affine transform without rotation. From training data in the supervised path, for each existing instance, we can calculate the affine transformation matrix $A$, which maps a box onto the object. Furthermore, we learn a neural network $G_l$, shared by both paths, which predicts $\hat{A}$ conditioned on $\mathbf{x}$, so that the preferred locations are determined according to the global context of the input. As such, we aim to find a realistic transform $\hat{A}$ which gives a result that is indistinguishable from the result of $A$. We use two discriminators; $D_{layout}^{box}$ which focuses on finding whether the new bounding box fits into the layout of the input semantic map, and $D_{affine}$ which aims to distinguish whether the transformaion parameters are realistic. Let $D_l$ denote the above discriminators that are related to the location prediction. Then, a minimax game between $G_l$ and $D_l$ is formulated as $\min_{G_l} \max_{D_l} \mathcal{L}_l(G_l, D_l)$. We consider three terms for the objective $\mathcal{L}_l$ as follows:

$$\mathcal{L}_l(G_l, D_l) = \mathcal{L}_l^{adv}(G_l, D_{layout}^{box}) + \mathcal{L}_l^{recon}(G_l) + \mathcal{L}_l^{sup}(G_l, D_{affine}), \tag{1}$$

where the first term is an adversarial loss for the overall layout, and other two terms are designed to regularize $G_l$. We visualize each term in Figure 2 with red arrows.

**Adversarial layout loss $\mathcal{L}_l^{adv}(G_l, D_{layout}^{box})$.** For an unsupervised path in Figure 2, we first sample $z_l \sim \mathcal{N}(\mathbf{0}, \mathbf{I})$ for an input. Information of $\mathbf{x}$ and $z_l$ is encoded to a vector $e$ and fed to a spatial transformer network to predict an affine transform $\hat{A}$. Finally, a new semantic map is generated by composing a transformed box onto the input. An adversarial loss for the unsupervised part is[2]

$$\mathcal{L}_l^{adv}(G_l, D_{layout}^{box}) = \mathbb{E}_{\mathbf{x}}[\log D_{layout}^{box}(\mathbf{x} \oplus A(b))] + \mathbb{E}_{\mathbf{x},z_l}[\log(1 - D_{layout}^{box}(\mathbf{x} \oplus \hat{A}(b)))]. \tag{2}$$

**Input reconstruction loss $\mathcal{L}_l^{recon}(G_l)$.** Although the adversarial loss aims to model the distribution of objects in the data, it frequently collapses to a few number of modes instead of covering the entire distribution [7]. For example, the first heatmap in Figure 2(b) presents the predicted bounding boxes using 100 different samples of $z_l$. It shows that the inferred location from $\hat{A}$ is almost the same for different random vectors. As a remedy, we reconstruct $\mathbf{x}$ and $z_l$ from $e$ to make sure that both are encoded in $e$. We add new branches at $e$ for reconstruction and train the network again with (2) and the following loss:

$$\mathcal{L}_l^{recon}(G_l) = \|\mathbf{x}' - \mathbf{x}\|_1 + \|z_l' - z_l\|_1, \tag{3}$$

where $\mathbf{x}'$ and $z_l'$ represent reconstructions. However, the generated bounding boxes are still concentrated at a few locations. To alleviate this problem, we use supervision that can help to find a mapping between $z_l$ and $\hat{A}$.

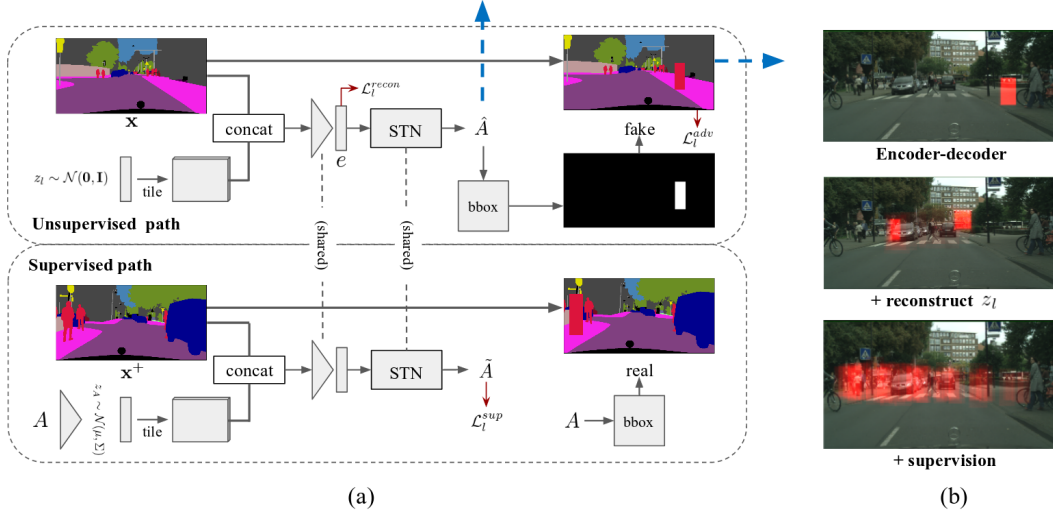

(a)                                                          (b)

Figure 2: (a) Network architecture of the *where* module. Blue arrows indicate connections with the *what* module in Figure 3. Red arrows denote the three loss terms in (1). (b) The learned context-aware spatial distributions of inserting a new person into the input semantic map $\mathbf{x}$. The distributions are shown as the heatmaps on images by sampling 100 different random vectors $z_l$. The top shows the mode collapse issue if we only use the adversarial loss $\mathcal{L}_l^{adv}$. The middle shows, by adding the reconstruction loss $\mathcal{L}_l^{recon}$ in training, it alleviates the mode collapse issue. The bottom shows, by further adding the supervised loss $\mathcal{L}_l^{sup}$ in training, the learned distribution becomes more diverse.

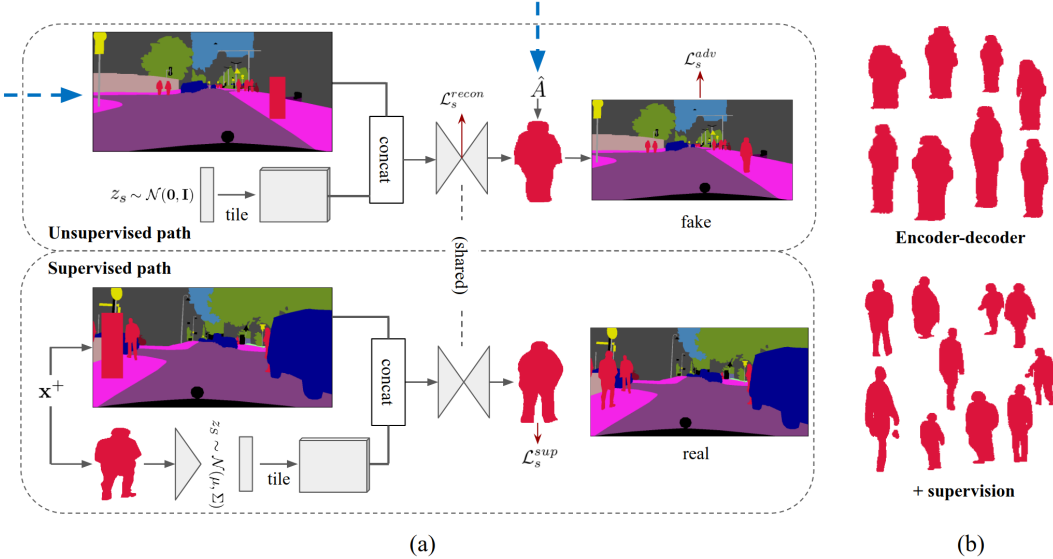

(a)                                                          (b)

Figure 3: (a) Network architecture of the *what* module, which learns the distribution of plausible shape, pose, and scale of object instances. Blue arrows indicate connections with the *where* module in Figure 2. Red arrows denote the three loss terms in (5). (b) Samples of the learned object instances (i.e., person) to be inserted into the input semantic map $\mathbf{x}$. Similar to Figure 2, the top shows the mode collapse issue if we only use the adversarial loss $\mathcal{L}_s^{adv}$ (and the reconstruction loss $\mathcal{L}_s^{recon}$). The bottom shows, by adding the supervised loss $\mathcal{L}_s^{sup}$, the learned distribution becomes multimodal.

**VAE-Adversarial loss $\mathcal{L}_l^{sup}(G_l, D_{affine})$.** In the supervision path, given $\mathbf{x}^+$, $A$ is one of affine transforms that makes a new realistic semantic map. Therefore, $G_l$ should be able to predict $A$ based on $\mathbf{x}^+$ and $z_A$ which is an encoded vector from parameters of $A$. We denote $\tilde{A}$ as the predicted transform from supervision as shown in Figure 2. The objective for the supervised path combines a

VAE and an adversarial loss [12]:

$$\mathcal{L}_l^{sup}(G_l, D_{affine}) = \mathbb{E}_{z_A \sim E_A(A)} \|A - \tilde{A}\|_1 + KL(z_A\|z_l) + \mathcal{L}_l^{sup,adv}(G_l, D_{affine}), \quad (4)$$

where $E_A$ is an encoder that encodes parameters of an input affine transform, $KL(\cdot)$ is the Kullback-Leibler divergence, and $\mathcal{L}_l^{sup,adv}$ is an adversarial loss that focuses on predicting a realistic $\tilde{A}$. Since the objective asks $G_l$ to map $z_A$ to $A$ for each instance, the position determined by the transform becomes more diverse, as shown in Figure 2(b).

As predicting a location of a bounding box mostly depends on the structure of the scene, so far we use a low-resolution input, e.g., $128 \times 256$ pixels, for efficiency. Note that with the same $\hat{A}$, we can transform a box to a high-resolution map, e.g., $512 \times 1024$ pixels, for more sophisticated tasks such as a shape generation in the following section.

### 3.2   The *what* module: learning a shape distribution of object instances

As shown in Figure 3, given the input semantic map $\mathbf{x}$, the *what* module aims to learn the shape distribution of object instances, with which the inserted object instance fits naturally within the surrounding context. Note that the shape, denoting the instance mask, also contains the pose information, e.g., a car should not be perpendicular to the road. The input to the generator network $G_s$ is a semantic map $\mathbf{x}$ with a bounding box $\hat{A}(b)$ (the output from the *where* module), and a random vector $z_s$, while the output is a binary mask of the instance shape $s$, i.e., $G_s(\mathbf{x} \oplus \hat{A}(b), z_s) = s$. Similar to the location prediction network, as shown in Figure 3(a), we set a minmax game between $G_s$ and discriminators $D_s$ as $\min_{G_s} \max_{D_s} \mathcal{L}_s(G_s, D_s)$ where $D_s$ consists of $D_{layout}^{instance}$, which checks whether the new instance fits into the entire scene, and $D_{shape}$, which examines whether the generated shape of the object $s$ is real or fake. There are three terms in $\mathcal{L}_s(G_s, D_s)$:

$$\mathcal{L}_s(G_s, D_s) = \mathcal{L}_s^{adv}(G_s, D_{layout}^{instance}) + \mathcal{L}_s^{recon}(G_s) + \mathcal{L}_s^{sup}(G_s, D_{shape}). \quad (5)$$

The role of each term is similar to (1) except that $\mathcal{L}_s^{sup}$ aims to reconstruct the input shape instead of transformation parameters. Figure 3(b) shows that the supervised path plays an important role to generate diverse shapes.

### 3.3   The complete pipeline

The *where* and *what* modules are connected by two links: first, the input to the unsupervised path of the *what* module is the generated layout from the *where* module; second, we apply the same affine transformation parameters to the generated shape, so that it can be inserted to the same location as being predicted in the *where* module. Therefore, the final output is obtained as follows:

$$\hat{\mathbf{x}} = \mathbf{x} \oplus \hat{A}(s), \quad (6)$$

where $\hat{A}$ and $s$ are generated from the generator of the *where* and the *what* modules, respectively:

$$\hat{A} = G_l(\mathbf{x}, z_l), \quad s = G_s(\mathbf{x} \oplus \hat{A}(b), z_s). \quad (7)$$

The STN is able to make a global connection between $\mathbf{x}$ from the input end, and $\hat{\mathbf{x}}$ in the output end. In addition, $\mathcal{L}_s^{adv}$ checks the fidelity of the $\hat{\mathbf{x}}$ in a global view. The complete pipeline enables that all the encoders to be adjusted by the loss functions that either lies in different modules, or with the global score. It potentially benefits the generation of many complicated cases, e.g., objects with occlusions.

## 4   Experimental Results

For all experiments, the network architecture, parameters, and initialization basically follow DCGAN [20]. For $D_{layout}^{box}$ and $D_{layout}^{instance}$, we use a PatchGAN-style discriminator. For $D_{affine}$, a fully connected layer maps 6-dimensional input to 64 and the other fully connected layer shrinks it to one dimension. For the *where* module, we place the bounding box in a $128 \times 256$ pixels semantic map. Then, a $128 \times 128$ pixels instance is generated based on a $512 \times 1024$ pixels semantic map. We use transposed convolutional layers with 32 as a base number of filters to generate the shape, while we use 16 for convolutional layers in the discriminator. The batch size is set to 1 and instance normalization is used instead of batch normalization.

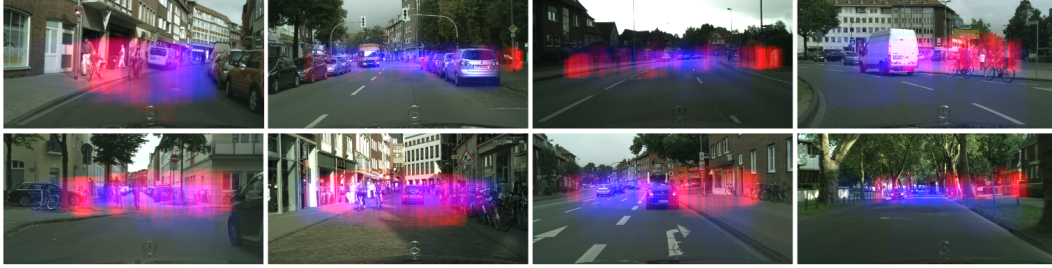

Figure 4: The learned spatial distributions (i.e., plausible locations and scales) of inserting persons (shown as red) and cars (shown as blue) for different input images. The distributions are shown as the heatmaps, by sampling input random vectors $z_l$ in the *where* module.

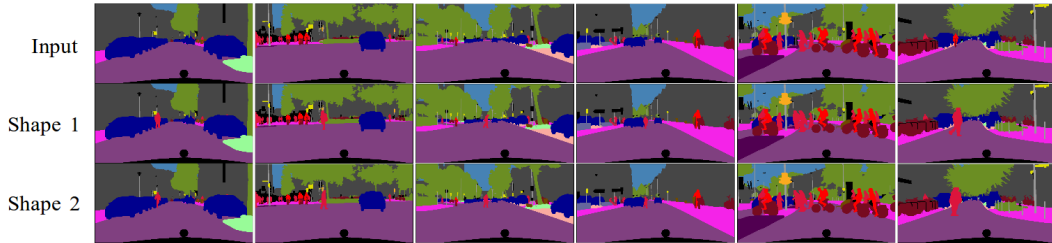

Figure 5: Effect of the random vector $z_s$ on shape generation from the *what* module.

**Layout prediction.** Figure 4 shows predicted locations for a person (red) and a car (blue) by sampling 100 different random vectors $z_l$ for each class. It shows that the proposed network learns a different distribution for each object category conditioned on the scene, i.e., a person tends to be on a sidewalk and a car is usually on a road. On the other hand, we show in Figure 5 that while fixing the locations, i.e., the *where* module, and by applying different $z_s$ in the *what* module, the network is able to generate object shapes with obvious variations.

**Baseline models.** The baseline models are summarized in Figure 6. For baseline 1, given an input scene, it directly generates an instance to a binary map. Then, the binary map is composed with an input map to add a new object. We apply real/fake discriminators for both the binary map and the resulting semantic map. While this baseline model makes sense, it fails to generate meaningful shapes, as shown in Figure 7. We attribute this to the huge search space. As the search space for where and what are entangled, it is difficult to obtain meaningful gradients. It indicates that decomposing the problem to where and what is crucial for the task.

The second baseline model decomposes the problem into *what* and *where* – an opposite order compared to the proposed method. It first generates an instance shape from the input and then finds an affine transform that can put both the object shape and a box on the input properly. As shown in results, the generated instances are not reasonable. In addition, we observe that STN becomes unstable while handling all kinds of shapes and then it causes a collapse to the instance generation network as well – the training is unstable. We attribute this to the fact that the number of possible object shapes given an input scene is huge. In contrast, if we predict the location first as proposed, then there is a smaller solution space that will significantly regularize the diversity of the object shapes. For example, a generated car have a reasonable front or a rear view when it is located at a straight lane. The results imply that the order of *where* and *what* is important for a joint training scheme.

**Ablation studies.** As discussed in (1) and (5), the input reconstruction loss and the VAE-adversarial losses are used to train the network. Figure 7 shows the results when the network is trained without each loss. It shows that the location and shape of the object are almost fixed for different images, which indicates that both losses are helping each other to learn a diverse mapping.

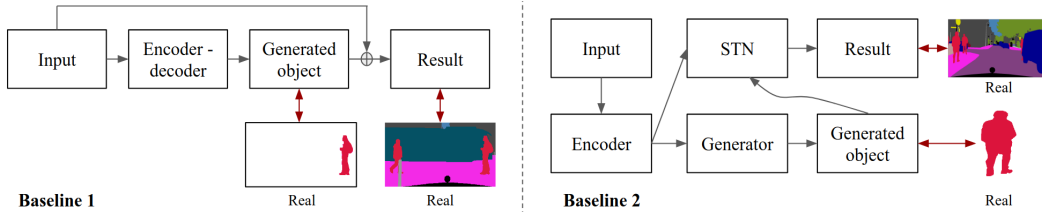

Figure 6: Two baseline architectures. Red arrows denote adversarial loss terms with real data.

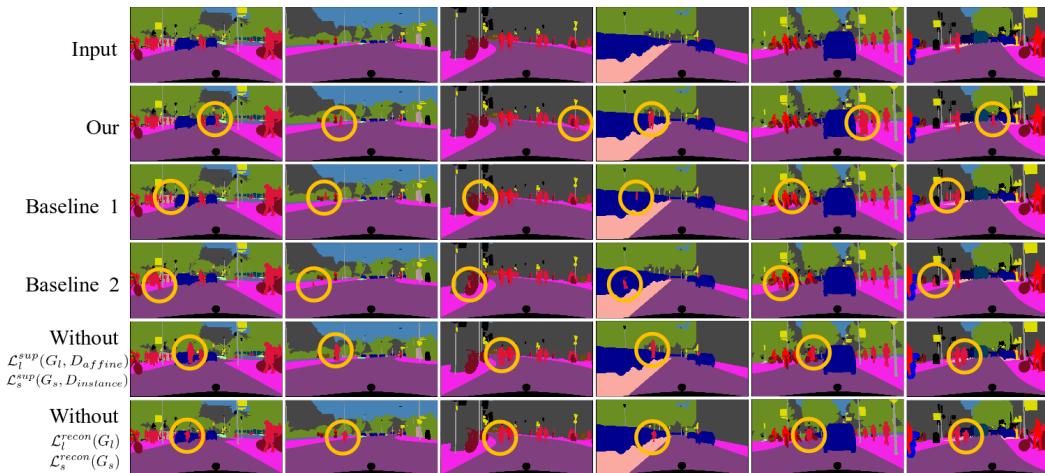

Figure 7: Results compared with the baseline methods. For a clear comparison, generated objects are marked with yellow circles. As shown, both baseline methods fail to generate realistic shapes of object instances. Our method (2nd row) synthesizes realistic new semantic maps by inserting new objects. The last two rows show the mode collapse issue, i.e., without either the reconstruction loss or the supervised loss, the predicted location and shape of the objects are almost fixed for different input images. Zoom-in to see details.

**Human evaluation.** We perform a human subjective test for evaluating the realism of synthesized and inserted object instances. A set of 30 AB test questions are composed using Amazon Mechanical Turk platform for the evaluation. In each question, a worker is presented two semantic segmentation masks. We put a marker above an object instance in each mask. In one mask, the marker is put on top of a real object instance. In the other, the marker is put on top of a fake object instance. The worker is asked to choose which object instance is real. Hence, the better the proposed approach, the closer the preference rate is to 50%. To ensure the human evaluation quality, a worker must have a lifetime task approval rate greater than 98% to participate in our evaluation. For each question, we gather answers from 20 different workers.

We find that for 43% of the time that a worker chooses the object instance synthesized and inserted by our approach as the real one instead of a real object instance in the original image. This shows that our approach is capable of performing the object synthesis and insertion task.

**Quantitative evaluation.** To further validate the consistency of layout distribution between the ground truth and inserted instances by our method, we evaluate whether RGB images rendered using the inserted masks can be reasonably recognized by a state-of-the-art image recognizer, e.g., YOLOv3 [21]. Specifically, given an RGB image generated by [31], we check whether the YOLOv3 detector is able to detect the inserted instance in the rendered image as shown in Figure 8 and Figure 9. As the state-of-the-art detector utilizes context information for detecting objects, we expect its detection performance will degrade if objects are inserted into semantically implausible places. We use the pretrained YOLOv3 detector with a fixed detection threshold for all experiments. Thus, the recall of the detector for new instances indicates whether the instances are well-inserted. We compare with four baseline models that are trained without one of the proposed discriminators. Table 2 shows that the proposed algorithm performs best when all discriminators are used.

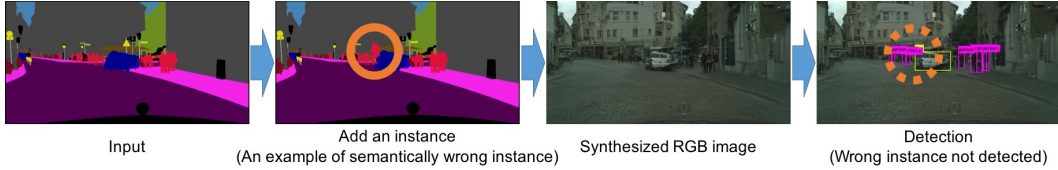

| Input | Add an instance (An example of semantically wrong instance) | Synthesized RGB image | Detection (Wrong instance not detected) |

Figure 8: For a quantitative evaluation of a semantic map, we first synthesize an RGB image and then determine whether the new instance is properly rendered in the image based on a detector.

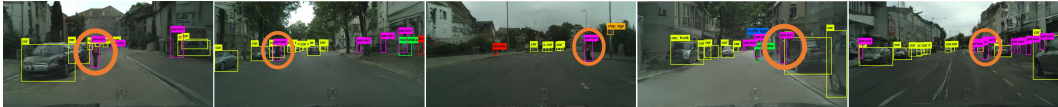

Figure 9: Synthetic images on top of manipulated segmentation maps via the proposed algorithm and detection results using YOLOv3. The generated objects are enclosed with yellow circles.

Table 2: Recall of YOLOv3 to detect an added person on the Cityscape dataset.

|  | wo $D_{layout}^{box}$ | wo $D_{affine}$ | wo $D_{layout}^{instance}$ | wo $D_{shape}$ | Full model |
| --- | --- | --- | --- | --- | --- |
| Recall | 0.60 | 0.70 | 0.68 | 0.71 | **0.79** |

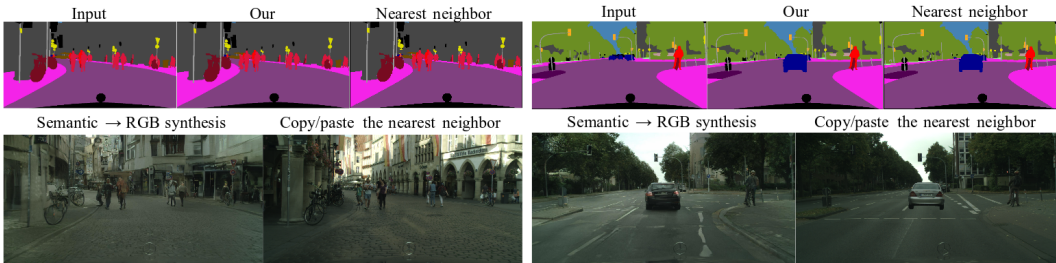

Figure 10: Samples of obtained RGB images from the synthesized semantic map.

**Applications: two ways to synthesize new images.** Our framework is flexible to be utilized – we can synthesize an RGB image in two different ways. First, an image can be rendered using an image synthesis algorithm which takes a semantic map as an input, such as [31]. The other way is finding the nearest neighbor of the generated shape. Then, we can crop the corresponding RGB pixels of the neighbor and paste it onto the predicted mask. Figure 10 shows that both approaches are working on the generated semantic map. It suggests that based on the generated semantic map, the new object can be well fitted into a provided image.

## 5 Conclusion

In this paper, we construct an end-to-end trainable neural network that can sample the plausible locations and shapes for inserting an object mask into a segmentation label map, from their joint distribution conditioned on the semantic label map. The framework contains two parts to model where the object should appear and what the shape should be, using two modules that are combined with differentiable links. We insert instances on top of semantic layout instead of image domain because the semantic label map space that are more regularized, more flexible for real-world applications. As our method jointly models **where** and **what**, it could be used for solving other computer vision problems. One of the interesting future works would be handling occlusions between objects.

**Acknowledgements** This work was conducted in NVIDIA. Ming-Hsuan Yang is supported in part by the NSF CAREER Grant #1149783 and gifts from NVIDIA.

## Footnotes

[2]We denote $\mathbb{E}_{(\cdot)} \triangleq \mathbb{E}_{(\cdot) \sim p_{data}(\cdot)}$ for notational simplicity, $input \oplus mask$ denotes blending the mask into the input. For example, $\mathbf{x}^+ \oplus A(b)$ and $\mathbf{x} \oplus \hat{A}(b)$ are a pair of real/fake masks in Figure 2(a).

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
