[Supplementary Material]

# Context-Aware Synthesis and Placement of Object Instances
# Supplementary Material

**Donghoon Lee**[1,2]*,     **Sifei Liu**[3],     **Jinwei Gu**[3],     **Ming-Yu Liu**[3],
**Ming-Hsuan Yang**[2,4],     **Jan Kautz**[3]
donghoon.lee@rllab.snu.ac.kr   {sifeil, jinweig, mingyul}@nvidia.com
mhyang@ucmerced.edu   jkautz@nvidia.com
[1]Seoul National University
[2]Google Cloud AI
[3]NVIDIA
[4]University of California at Merced

In this supplementary material, we describe additional experimental results.

**GIF files.**    In addition to the pdf document, we provide six gif files in the supplementary material. File one to four show training progress of the *where* network from epoch 0 to 30. It shows learned spatial distributions of inserting persons (shown as red) and cars (shown as blue). The distributions are shown as the heatmaps, by sampling input random vectors $z_l$ in the *where* module. As the epoch increases, the predicted distribution becomes more diverse and reasonable.

For file five and six, we sample different random vectors $z_s$ in the *what* module while $z_l$ is fixed. The results show that the proposed algorithm renders diverse and realistic person shapes at a fixed location, i.e., the effect of $z_l$ and $z_s$ are disentangled.

**Manipulated semantic maps.**    Figure 1, Figure 2 and Figure 3 show results of the proposed algorithm. For each semantic map, we inserted a single person.

Figure 1: Results of an inserted person in the semantic map.

Figure 2: Results of an inserted person in the semantic map.

Figure 3: Results of an inserted person in the semantic map.

## Footnotes

*This work is mainly done when Donghoon Lee was an intern at NVIDIA.