[Reviews · NeurIPS 2018]

Reviewer 1



The paper is concentrated in a new task in which we want to model the placement of object given its semantics and context. The proposed method disentangles the generation process of possible positions and possible shapes, which is certainly a plus. STN serves as a bottleneck between two modules and made the whole network end-to-end trainable. At the same time, there are some issues in the current submission: 1. Missing training details The model consists of two generators and four discriminators. To be honest, I'm doubtful about the training procedure itself, especially no specific details given, such as whether there is mode-dropping or not. 2. Missing qualitative evaluation Even though it's imaginable that for this new task, no direct qualitative metrics could be applied, this part is a big minus to the convincibility of the proposed method for its audience. 3. Missing ablation study The paper also misses a solid qualitative ablation study. As is, there might be some visual differences between the presented design and ablated design in Fig. 5. But the differences are minor and not clear enough to be considered as non-trivial improvement or necessity, especially given a very condense and bulky network architecture. Overall, this is a paper on the right problem but w/o convincing deliberation of design or qualitative results.

Reviewer 2



In overall, I think this paper proposes a well-designed pipeline for semantic map synthesis which is helpful to image generation tasks. The rebuttal addresses my questions about the ablation study. I strongly suggest the authors to add the ablation study in the final version if this paper is accepted. My final rating is accept. ========================================= This paper proposes a where and what pipeline to synthesize and place object instances into a semantic map in an end-to-end way. This method can help to generate diverse natural semantic map which contains important structure information. The generated semantic maps can benefit the image synthesis task. Pros: * The proposed idea is interesting that using the context information to predict the position and shape of object instances in order to generate novel semantic maps. * The writing is easy to follow, and empirical experiments show that it is able to synthesize natural semantic maps. * The decomposed where and what modules make scenes and can provide some control during the synthesis. The well-designed jointly training strategy is suitable for the semantic map synthesis task. The proposed framework can also be useful for other structure generation tasks. Cons: * In total, there are four discriminators. I wonder how do they affect the training and results. Is there any ablation study for this? Given that this paper is about the context-aware location and shape prediction, it is suggested that some recent work on context based structure prediction for image inpainting and generation can be included in the related work, e.g., [a] Q. Sun, L. Ma, S.J. Oh, L. Van Gool, B. Schiele, and M. Fritz. “Natural and Effective Obfuscation by Head Inpainting.” In CVPR, 2018. [b] K. Ehsani, R. Mottaghi, and A. Farhadi. “Segan: Segmenting and generating the invisible.” In CVPR 2018.

Reviewer 3



This is a well written paper that addresses the problem of inserting new objects into an image at plausible locations and scales. The proposed method uses a semantic map to learn so-called "where" and "what" models. The models are not particularly novel, employing conditional GANs but it is interesting how they work together as well as the introduction of the spatial transformer network. The main weaknesses that the paper does not address the issue of photometric consistency (unless this is somehow automatically handled by the GANs). For example, how does the model ensure that the correct lighting, shading and shadows are produced? The second weakness is that the model is only tested on the cityscapes dataset, which appears to be relatively simple for this task in that the scenes are relatively constrained and the camera viewpoint is fixed. Nevertheless, the paper does not overstate its contribution claiming to "take a first step to address the [...] problem."